# Effect of Dietary Phosphorous Restriction on Fibroblast Growth 2 Factor-23 and sKlotho Levels in Patients with Stages 1–2 Chronic Kidney Disease

**DOI:** 10.3390/nu14163302

**Published:** 2022-08-12

**Authors:** Anita Saxena, Trisha Sachan, Amit Gupta, Vishwas Kapoor

**Affiliations:** 1Department of Nephrology, Sanjay Gandhi Post Graduate Institute of Medical Sciences, Lucknow 226014, India; 2Apollo Medics Kanpur Road, Lucknow 226012, India; 3Department of Biostatistics and Bioinformatics, SGPGIMS, Lucknow 226014, India

**Keywords:** chronic kidney disease, hyperphosphatemia, fibroblast growth factor-23, dietary phosphorus intake, dietary intervention

## Abstract

Hyperphosphatemia has emerged as an independent risk factor for cardiovascular disease (CVD) and excess mortality in chronic kidney disease (CKD). The study evaluates the effect of dietary phosphorus (Ph) restriction (DPhR) at an early stage as a therapeutic strategy for delaying CKD progression and preventing CVD. Methods: This was a one-year interventional study conducted on 79 stage 1 and 2 CKD patients. The dietary phosphorus intake (DPhI), fibroblast growth factor-23 (FGF-23), sKlotho and serum phosphorous (SP) levels were analyzed. Patients were categorized into two groups based on their DPhI, recommended DPhI (RPhI) with <1000 mg/day of dietary phosphorous (dietary counselling) and high DPhI (HPhI) with >1000 mg/day (dietary intervention). For comparisons of differences between the two groups, independent *t*-test; for correlation analysis, Pearson correlation; for identifying the significant associated risk factors for CKD, binary logistic regression analysis and for comparing the means across the three visits, repeated measures ANOVA were used for statistical analysis. Results: The mean age and glomerular filtration rate (GFR) of CKD patients were 38 ± 12 years and 82.95 ± 16.93 mL/min/1.73 m^2^. FGF-23, SP, dietary protein and DPhI were significantly higher and sKlotho was significantly lower in HPhI group than RPhI group. In HPhI group; GFR, sKlotho, SP and FGF-23 correlated significantly with DPhI. Risk factors with a statistical bearing on the progression of CKD were animal-based diet, family history of CKD and hypertension. In HPhI group; GFR, DPhI, SP and FGF-23 levels significantly improved within the intervention period whereas a significant increase in sKlotho levels was observed in both the groups. Conclusion: Restricting DPhI emerged as a favorable therapeutic strategy for CKD patients for improving renal function and controlling hyperphosphatemia. The results of the present study may serve as the basis for future interventional studies with dietary phosphate restriction in the initial stages of CKD that would preserve renal function. Highlights: Early restriction of dietary phosphorus prevents decline in eGFR, elevation in FGF23 and increases Klotho levels.

## 1. Introduction

Chronic kidney disease (CKD) has become a worldwide public health problem that should be managed in its early stages [1]. For this, it is important to identify risk factors associated with increased risk of renal disease. The traditional risk factors for the progression of renal disease include uncontrolled diabetes, hypertension, a family history of CKD, and older age [2]. In the last two decades, several observational studies advocated hyperphosphatemia as an emerging independent cardiovascular risk factor in CKD [3] which is known to be associated with mineral and bone disorder (MBD), cardiovascular disease (CVD) [4], and worsening of kidney function [5] contributing to an increase in mortality of CKD patients [6].

Appropriate levels of phosphate in the body are maintained by the coordinated regulation of the bone-derived phosphaturic hormone, fibroblast growth factor-23 (FGF-23) and most of its functions occur in the kidney through an FGF receptor (FGF-R)/Klotho complex [7].

High phosphorous load increases serum FGF-23 levels in the early stages of CKD, which inhibits renal 1α-hydroxylase thus inhibiting calcitriol synthesis and reducing gut absorption of calcium which causes hypocalcemia. Hypocalcemia, in turn, augments increased production of parathyroid hormone (PTH) [3,8], thus leading to secondary hyperparathyroidism [9].

Hence, it is important to monitor dietary phosphorus intake (DPhI) of patients with renal insufficiency, protein is the primary source of dietary phosphorus. According to KDOQI guidelines, phosphorous-to-protein ratio (PPR) is not only an important culprit of CKD-MBD but has also been reported to be related to increased mortality, and therefore calls for greater monitoring of the same [10,11]. Therefore, dietary phosphorus restriction (DPhR) should be advised to CKD patients at an early stage as a therapeutic strategy for delaying progression of CKD and preventing cardiovascular disease. To date, the impact on FGF-23 in manipulating DPhI in CKD patients has not been studied much in detail. With this background we conducted a study to elucidate the intake of dietary phosphorus and PPR and the association of the two with markers of renal function in CKD patients and analyze effect of dietary intervention on the levels of FGF-23, sKlotho, serum phosphorous, and biochemical parameters of CKD patients at 6 and 12 months after initiating intervention.

We hypothesized that higher dietary phosphorus intake and intake of foods with higher phosphorous-to-protein ratio may accelerate a decline in renal function which might improve with the tightly controlled DPhI in patients with CKD stages 1–2.

## 2. Methods

This was a longitudinal study, approved by the Ethics Committee of the institute. Patients were recruited from outpatient departments of nephrology and endocrinology of the institute. The ethics approval code is IEC code 2015-116-IMP-87 Dated 05-06-2016. Informed consent was obtained from all the patients.

The inclusion criteria were (a) patients ≥ 18 years of age, and (b) eGFR of >90 (CKD stage 1) and 60–89 mL/min/1.73 m^2^ (CKD stage 2) [12].

The exclusion criteria were (1) patient should not have been on phosphate binders and (2) patient should not have received dietary counselling.

Estimated GFR (eGFR) was calculated using the short MDRD formula [13]. CKD-EPI equation was not used as the short MDRD formula was recommended by experts in the research/doctoral committee hence we had to comply by the recommendations of the research committee.

This study presents data on a total of 79 CKD patients. To find the association of FGF-23, Klotho, PTH levels with dietary phosphorus in CKD Stages 1 and 2 sample sizes was calculated in consultation with a bio-statistician. According to the null hypothesis, when r = 0, there is no correlation. A negative correlation will be at r = 0.5 and positive > 0.5 at a 5% alpha error and power of 90%, a minimum sample of 73 is required. Accordingly, for this study, 79 patients with CKD were recruited to ascertain the association.

Demographic data such as age, gender, blood pressure, diet type i.e., vegetarian (plant-based diet) or non-vegetarian (meat-based diet preferably taking 4–5 times a week); a medical history of the patient; biochemical parameters including eGFR, serum creatinine (using Jaffe’s method), serum phosphorous, iPTH, urinary phosphorous, urinary protein, serum calcium and vitamin D; and dietary data including dietary energy, dietary protein and phosphorous intake were collected. PPR of food items was calculated by dividing DPhI (mg/day) to dietary protein (gm/day).

Four milliliter of fasting blood sample was taken from all CKD patients for analyzing FGF23 (Kainos Lab Inc., Tokyo, Japan) [14] and sKlotho (Immuno-Biological Laboratories Co., Ltd. Tokyo, Japan) [15] levels which were assayed using an enzyme-linked immunosorbent assay (ELISA) technique as per the manufacturer’s protocol. 

Three days dietary recall was taken in a structured interview on their visit to the hospital. Average of the three days was taken for analysis. Nutrient analysis was done using Standard Nutrition Tables of National Institute of Nutrition (NIN) published by Indian Council of Medical Research (ICMR) [13].

Based on DPhI, patients were divided into two groups: recommended phosphorous intake (RPI < 1000 mg/day) and high phosphorous intake (HPhI > 1000 mg/day). The rationale to divide groups according to phosphorus intake < 1000 and ≥1000 mg/day was based on an average healthy Indian’s protein intake which is 0.6–0.7 g/kg/day (equal to approximately 700–800 mg/day of phosphorus intake daily). Hence, the groups were divided based on KDOQI guidelines for phosphorus intake in CKD, which is 800–1000 mg/day (although in CKD stages 1 or 2, neither phosphorus restriction nor use of phosphate binders is recommended).

Recommended phosphorous intake (RPhI) group: This group was defined as recommended DPhI of <1000 mg/day; without any previous counselling or dietary intervention. These patients were given dietary counselling only on their hospital visits and telephonically for motivating them to follow the diet plan in between, and high phosphorous intake (HPhI) group patients with high DPhI i.e., >1000 mg/day were included in this group. This group underwent a dietary intervention which included intense dietary counselling, prescribed dietary modifications and individualized tailored diet plans based on their age, weight, lifestyle, and likes and dislikes toward the food items. The patients were counselled to adopt plant-based diet. This group underwent focused dietary intervention which included intense dietary counselling, prescribing dietary modifications, and individualized tailored diet plans given and prepared by a renal dietitian focusing on eliminate/restricting foods high in phosphorus and PPR based on their age, weight, lifestyle cultural fads, and likes and dislikes toward the food items. The patients were counselled to adopt a plant-based diet and milk and milk-based products. Patients were advised to eliminate all nuts and seeds, egg yolk, animal meat [16], bakery products, chocolates, packaged food, phosphorus-containing additives (cola beverages, processed meat, processed cheese) from their diet. Lentils and pulses, milk and milk products (hard cheese, ice-creams, custards, cottage cheese, pudding, yoghurt), bran and whole wheat cereals were restricted up to 1–2 servings a day as per patient’s health and the geographical area where the patient was residing as the mineral content of food varies with the mineral content of the soil of the particular area. Patients were also advised to prepare phosphorus-rich foods by wet cooking methods such as boiling, thereby, reducing phosphorus as well as sodium and potassium content while preserving nitrogen content of protein, reducing the effective phosphate intake per gram of dietary protein, allowing better control of phosphate balance together with a lower risk of protein malnutrition. Dietary adherences were monitored twice a month in all the patients. Compliance with the diet was ascertained through interviews conducted by the renal dietician using the detailed dietary questionnaire on the patient’s visit to the hospital.

Intake of phosphorous rich foods items was restricted and replaced with low phosphorus alternatives to meet phosphorous restriction in the diet plan. All the patients of HPhI group were prescribed at least 30 kcal/kg/day of energy with low dietary protein intake 0.6 g/kg/day and restricted phosphorous intake of 800–1000 mg/day [10] during the intervention period i.e., from baseline to twelve months. Although phyotochemicals and antioxidants have a bearing on phosphate metabolism, but this aspect was not evaluated.

Statistical analyses were performed using IBM, SPSS Statistics (University of Chicago, Chicago, IL, USA) for Windows, version 20. Normality of data was examined by Shapiro Wilk test. Results were presented as mean ± SD for normally distributed while median (IQR) for skewed data and categorical as n (%). For comparisons of differences between the two groups, bifurcated by the dietary phosphorous intake of the patients, chi-square test for categorical variables and independent *t*-test or Mann–Whitney U test for continuous variables was used. For correlation analysis, Pearson correlation was used, reported as correlation coefficient, r. For identifying the significant associated risk factors for CKD, odds ratios (OR) with 95% confidence interval (CI) were calculated using binary logistic regression analysis. Repeated measures ANOVA was used to compare the means across the three visits i.e., baseline, six, and twelve months of both the groups. The Greenhouse–Geisser correction was used when Mauchly’s test of sphericity was violated. Two-tailed *p*-values < 0.05 were considered statistically significant. We have represented data as mean +/− SD as it describes the dispersion of data points about mean which helps in understanding the distribution of data hence mean +/− SEM which describes the sampling distribution of sample mean was not used.

## 3. Results

Clinical characteristics of CKD patients are given in Table 1 and Table 2. The mean age of CKD patients was 38 ± 12 years with the mean eGFR of 82.9 ± 16.9 mL/min/1.73 m^2^.

Table 3 shows the biochemical and dietary data of CKD patients. CKD patients had high phosphorous intake (1104.64 ± 248.53 mg/day) and higher PPR (25.36 ± 3.95).

Further, we spilt DPhI of patients into two groups; RPhI and HPhI group to look for the association of phosphorus load on these biochemical parameters separately. Diet type (*p* = 0.042) categorized as vegetarian and non-vegetarian was found to be statistically different between the two groups. In both RPhI and HPhI groups, the serum phosphorous (*p* < 0.001) and iPTH (*p* = 0.040) levels were within normal range, though the mean values of patients in HPhI were higher when compared to those in the RPhI group and the difference was statistically significant. FGF-23 (*p* < 0.001) and urinary protein (*p* < 0.001) were significantly higher in HPhI than RPhI. sKlotho was significantly low (*p* < 0.001) in the HPhI group compared to the RPhI group.

eGFR of patients in HPhI group was lower than those of RPhI but the difference was not statistically significant. There was no statistical difference between groups in serum creatinine, urinary phosphorous, serum calcium, and Vitamin D.

Significantly higher intake of dietary protein (*p* < 0.001) and dietary phosphorous was (*p* < 0.001) observed in HPhI when compared with RPhI group. Because of the higher intake of protein and phosphorous by HPhI group, PPhR (*p* < 0.001) was found to be significantly higher when compared with the RPhI group. Energy intake of CKD patients between both groups was not statistically different (*p* = 0.380) (Table 4 and Table 5).

Univariable logistic regression analysis was performed to find out risk factors for CKD progression based on the dietary phosphorous intake of CKD patients (Table 6). Risk factors which had a statistical bearing on the progression of CKD were diet type, family history of CKD, and hypertension. Further on, multivariable logistic regression analysis was performed to find out risk factors for CKD progression (Table 7) and risk factors which had a statistical bearing on the progression of CKD were hypertension (AOR = 0.082, *p* value = 0.002) and family history of CKD (AOR = 2.895, *p* value = 0.047). Diet type was not significant in multivariable logistic regression analysis. However, there is a high likelihood of diet type (high phosphorus intake) having a bearing on progression in higher stages of CKD where serum phosphorus levels are overtly high. 

Univariable logistic regression analysis showed diet type, family history, and hypertension as risk factors for CKD progression in CKD patients.

Table 7 multivariable logistic regression analysis was performed to find out the risk factors for CKD progression based on phosphorus intake of CKD patients. Risk factors which had a statistical bearing on the progression of CKD were, hypertension (AOR = 0.082, *p* value = 0.002) and family history of CKD (AOR = 2.895, *p* value = 0.047).

In CKD patients, the effect of the dietary intervention over time was investigated using repeated measures of ANOVA. In HPhI group, eGFR (*p* = 0.012) significantly improved within the intervention period. There was a significant reduction in dietary phosphorous intake on the second and third visits of the patient at 6 and 12 months in the HPhI group. The reduction in DPhI became apparent with a concomitant significant reduction in the serum phosphorous levels (*p* = 0.001) from baseline (3.95 ± 0.50 mg/dL) to 6 months (3.36 ± 0.78 mg/dL) in HPhI group but remained stable for the next 6 months (3.36 ± 0.78 mg/dL). Following a reduction in dietary and serum phosphorus levels, a remarkable significant decline in FGF-23 level of HPhI group (*p* < 0.001) patients was observed in the intervention period. In CKD patients, significant increases in Klotho levels were observed in both the groups i.e., RPhI group (*p* = 0.002) and HPhI group (*p* < 0.001). (Table 8 and Table 9).

## 4. Discussion

High phosphorus intake is known to cause renal and vascular calcification and renal tubular injury and albuminuria. Dietary phosphorous intake has a linear relationship with dietary protein; occurring naturally in protein-rich foods [17]. In our CKD cohort, we found a strong correlation of DPhI with dietary protein. Similar findings were reported by Salomo et al. [18] and Noori et al. [11]. In the present study, we found that in the early stages of CKD (mean eGFR of 83.69 ± 17.37 mL/min/1.73 m^2^), serum phosphorous and PTH were in the normal range even with increasing serum levels of FGF-23 [19] and these findings are supported by the Chronic Renal Insufficiency Cohort study (2011) by Isakova et al. [20].

The average protein intake in Indian diet is 0.6 g/kg/day. In our study, though, mean daily protein intake was normal, DPhI was above the recommended range, as K/DOQI guidelines recommend up to 1000 mg/day DPhI for CKD patients [10].

Higher intake of dietary phosphorous correlated positively with serum phosphorous in our study. Studies have found that high DPhI increases serum phosphorus concentrations [21]. However, not only the amount of phosphorus intake is responsible but also the type and source of phosphorus intake are responsible for significantly increasing serum phosphorus concentration. Moore et al. [22] performed a clinical trial and their analysis showed that cereals and dairy products with inorganic phosphate additives significantly increased serum phosphorus concentration, despite being consumed less frequently than foods without phosphate additives [22].

In the early stage of CKD, we found FGF-23 was directly associated with high dietary phosphorus intake. Studies have shown that phosphorus load leads to a faster decline in renal function as the damaged kidney is not able to excrete the excess phosphorous which causes elevated FGF-23 levels [23].

In our study, a negative correlation was found between eGFR and Klotho and high phosphorus intake. This finding is supported by the results obtained by in vitro studies showing that gene and protein expression of renal α- Klotho was reduced in rats on high phosphorus diet [24].

Further in the study, we split CKD patients into two groups based on their DPhI to observe whether the groups and other factors were still independently associated with the risk of CKD. 

CKD patients with recommended phosphorus intake had decreased serum phosphorus [25] and FGF-23 levels were significantly higher in CKD patients with high phosphorus intake. Yoshikawa et al. found higher levels of FGF23 in the mice fed on high phosphate diet group than in the mice on low phosphate diet group, whereas renal *Klotho* mRNA expression levels were found to be lower in the mice fed on high phosphate diet group than in the mice on low phosphate diet group [26] which supported our findings on Klotho that decreased significantly in HPhI group when compared with RPhI group.

Most of our CKD patients who were non-vegetarian were in the higher HPhI group, which is due to their high phosphorus intake. The findings of the study emphasize that in CKD patients predominantly on animal-based diets, there is a need for cautious dietary phosphorus intake from early stages of CKD as the univariable analysis has shown dietary phosphorus as a risk factor for CKD progression. However, in multivariable analysis the dietary phosphorus consumption did not appear as a risk factor for CKD progression, rather only family history of CKD and hypertension were risk factors for CKD progression. While taking the dietary recall, the renal dietician should educate and counsel CKD patient on the sources of phosphorus, and type of diet the patient should prefer. Moreover, the importance of low phosphorus-to-protein-ratio diet should be emphasized in dietary counselling and educating patients with CKD [27].

Even though patients of RPhI group received only dietary counselling, there was a slight increase in eGFR but Klotho levels significantly improved; although dietary phosphorous intake of this group slightly increased over 12 months. In the strict diet counselling group, patients, phosphorous intake was within the recommended RDAs, i.e., below 1000 mg.

FGF-23 levels during the interventional period decreased significantly and came within the normal range. Furthermore, reduction in DPhI became apparent with a concomitant significant reduction in the serum phosphorous levels in HPhI group but remained within the normal range with improvement in eGFR (*p* = 0.012) within the intervention period implying, thereby, that controlled dietary phosphorous intake (i) might improve the renal function, (ii) it may not affect patient satisfaction in terms of food intake, and consequently, (iii) could stabilize biochemical parameters as well. It is clear from this study that those who received dietary counseling alone, even a slight increase in DPhI influenced FGF-23 and sKlotho levels implying, thereby, that strict adherence to low intake of dietary phosphorous should be initiated in CKD stage 1.

Conclusion: CKD patients should be assessed frequently to determine whether they are at increased risk of CKD progression. Family history of CKD and hypertension are risk factors for CKD progression. Animal/meat-based diet should be monitored. Elevated blood pressure should be monitored in CKD patients to delay CKD progression. There is a need to further develop approaches to ensure compliance to prescribed diets to CKD patients which should be carefully done in consultation with a renal dietician as undesirable intake of nutrients can accelerate deterioration of kidney function.

## Figures and Tables

**Table 1 nutrients-14-03302-t001:** Clinical characteristics of CKD patients (n = 79).

Number of Patients	79
Gender-	
Male/Female	42/37 *
42/37 *	
Veg	46
Male	23 (50%) *
Female	23 (50%) *
non-veg	33
Male	19 (57.6%) *
Female	14 (42.4%) *
Mean age, years	38 ± 12 ^†^
Male	41 ± 9 ^†^
Female	35 ± 13 ^†^
Glomerular filtration rate, mL/min/1.73 m^2^	83.6 ± 17.3 ^†^
Male	85.1 ± 17.9 ^†^
Female	82.0 ± 16.7 ^†^
Diabetes, number (%)	26 (32.9%) *
Male	13 (16.4%) *
Dyslipidemia	15 (18.9%) *
Hypertensive	23 (29.1%) *
Family history with CKD	37 (46.8%) *
Renal diagnosis	
Diabetic *kidney disease*	18 (22.7%) *
Polycystic kidney disease	14 (17.7%) *
Other renal diseases	16 (20.2%) *
Unknown	31 (39.2%) *
Medications	
Anti-hypertensive drug(s)	23 (29.1%) *
Statins	16 (20.2%) *
Insulin or oral hypoglycemic agent	26 (32.9%) *

Values are presented in mean ± SD/n (%). CKD: chronic kidney disease. ^†^ Continuous data are represented as mean ± SD. * Categorical data are represented as frequency (percentage).

**Table 2 nutrients-14-03302-t002:** Clinical characteristics of CKD patients (n = 79).

		Male (n = 42)	Female (n = 37)
Diet type-			
Veg	46	23 (29.11%)	23 (29.11%)
non-veg	33	19 (24.05%)	14 (17.72%)
Diabetes, number (%)	26 (32.9%)	13 (16.4%)	13 (16.4%)
Dyslipidemia	15 (18.9%)	6 (7.59%)	9 (11.39%)
Hypertensive	23 (29.1%)	12 (15.18%)	11 (13.92%)
Family history with CKD	37 (46.8%)	18 (22.78%)	19 (24.05%)
Mean age, years	38 ± 12 ^†^	41 ± 9 ^†^	35 ± 13 ^†^
Glomerular filtration rate, mL/min/1.73 m^2^	83.7 ± 17.4 ^†^	85.1 ± 18 ^†^	82.1 ± 16.8 ^†^
Renal diagnosis			
Diabetic kidney disease	18 (22.7%)	10 (12.65%)	8 (10.12%)
Polycystic kidney disease	14 (17.7%)	6 (7.59%)	8 (10.12%)
Other renal diseases	16 (20.2%)	9 (11.39%)	7 (8.86%)
Unknown	31 (39.2%)	17 (21.51%)	14 (17.72%)
Medications			
Anti-hypertensive drug(s)	23 (29.1%)	12 (15.18%)	11 (13.92%)
Statins	16 (20.2%)	7 (8.86%)	9 (11.39%)
Insulin or oral hypoglycemic agent	26 (32.9%)	13 (16.4%)	13 (16.4%)

Values are presented in mean ± SD/n (%). CKD: chronic kidney disease. ^†^ Data are represented as mean ± SD.

**Table 3 nutrients-14-03302-t003:** Biochemical and dietary data of CKD patients (n = 79).

Parameters (Unit) (n = 79)	Results
Serum creatinine (mg/dL)	0.9 (0.8–1.2) ^†^
Serum phosphorous (mg/dL)	3.6 (2.9–4.1) ^†^
iPTH (pg/mL)	51.8 (43.9–60.2) ^†^
FGF-23 (pg/mL)	56.7 (53.2–61.6) ^†^
sKlotho (pg/mL)	703.6 (658.5–831.4 ^†^)
Urinary phosphorous (mg/day) *	649 (408.3–804.3) ^†^
Urinary protein (mg/mL) #	20 (16.8–22.5) ^†^
Serum calcium (mg/dL)	8.79 ± 0.77 ^$^
Vitamin D (ng/mL)	26.30 ± 8.37 ^$^
Energy intake (kcal/kg/day)	26.54 ± 3.54 ^$^
Dietary Protein intake (g/day)	43.53 ± 7.42 ^$^
Dietary phosphorous (mg/day)	1104.64 ± 248.53 ^$^
Phosphorous to protein ratio	25.36 ± 3.95 ^$^

* n = 63. # n = 59. FGF-23: fibroblast growth factor-23; iPTH: intact parathyroid hormone values are presented in ^$^ mean ± SD and ^†^ median (Q1–Q3).

**Table 4 nutrients-14-03302-t004:** Demographic and baseline biochemical data of the CKD patients based on their dietary phosphorous intake (mg/day).

Parameters (Unit)	RPI (n = 37)	HPHI (n = 42)	*p*-Value
Diet type-			0.042 †
Vegetarian	26	20	
Non-vegetarian	11	22	
Gender-			0.882 †
Male	20	22	
Female	17	20	
GFR (mL/min/1.73 m^2^)	90.5 (67–99.7)	86.2 (65.1–93.7)	0.549 ^$^
Serum creatinine (mg/dL)	0.9 (0.8–1.2)	1 (0.8–1.2)	0.501 ^$^
Serum phosphorous (mg/dL)	3 (2.7–3.6)	3.8 (3.6–4.4)	<0.001 ^$^
iPTH (pg/mL)	45.5 (41.8–58.3)	52 (49.5–63.7)	0.011 ^$^
FGF-23 (pg/mL)	53.9 (51–57.2)	58.4 (54.4–68.2)	<0.001 ^$^
sKlotho (pg/mL)	831.4 (705–862.5)	673.9 (603.7–701)	<0.001 ^$^
Urinary phosphorous (mg/day) *	655.8 (476.6–797.4)	629.4 (399.7–804.3)	0.595 ^$^
Urinary protein (mg/mL) #	17.7 (15.7–19.9)	21 (19.8–24.1)	<0.001 ^$^
Serum calcium (mg/dL)	8.89 ± 0.60	8.70 ± 0.89	0.267 ^@^
Vitamin D (ng/mL)	25.72 ± 8.23	26.81 ± 8.56	0.566 ^@^
Energy intake (kcal/kg/day)	25.85 ± 3.26	27.15 ± 3.70	0.105 ^@^
Dietary Protein intake (g/day)	37.57 ± 3.40	48.79 ± 5.86	<0.001 ^@^
Dietary phosphorous (mg/day)	868.96 ± 69.99	1312.26 ± 137.57	<0.001 ^@^
Phosphorous to protein ratio	23.24 ± 2.18	27.22 ± 4.24	<0.001 ^@^

* n = 29 (RPI) and n = 34 (HPhI). # n = 28 (RPhI) and n = 31 (HPhI). FGF-23: fibroblast growth factor-23; iPTH: intact parathyroid hormone; GFR: glomerular filtration rate values are presented in mean ± SD and median (Q1–Q3). ^@^ *p* value was calculated using independent sample *t*-test; ^$^ *p* value was calculated by Mann Whiney U test; † *p*-value calculated by chi-square test.

**Table 5 nutrients-14-03302-t005:** Demographic and baseline biochemical data of the CKD patients based on gender.

	Male (n = 42)	Female (n = 37)	*p* Value
GFR	85.8 (67.6–99.7)	90.1 (64–93.7)	0.415
Serum creatinine (mg/dL)	1 (0.9–1.2)	0.8 (0.7–1.1)	<0.001
Serum phosphorous (mg/dL)	3.7 (3.2–4.1)	3.5 (2.8–3.8)	0.374
iPTH (pg/mL)	51.5 (43.3–60.1)	51.8 (44.7–60.2)	0.655
FGF-23 (pg/mL)	57.3 (53.6–61.6)	55.2 (52.2–59.1)	0.196
sKlotho (pg/mL)	702.8 (648.7–811.7)	703.6 (673.9–840.7)	0.482
Urinary phosphorous (mg/day) *	661 (476.6–798.6)	638.9 (403.4–804.3)	0.448
Urinary protein (mg/mL) #	20.5 (18.3–22.5)	19.4 (15.8–22.5)	0.124
Serum calcium (mg/dL)	8.9 (8.6–9.1)	9 (8.3–9.3)	0.555
Vitamin D (ng/mL)	28.1 (19.8–32.4)	27.4 (18.7–30.8)	0.613
Energy intake (kcal/kg/day)	10.2 (7.8–14.6)	11 (7.4–14.5)	0.804
Dietary Protein intake (g/day)	43.7 (38.7–52.1)	40.3 (36.6–47.9)	0.054
Dietary phosphorous (mg/day)	1161.7 (883.5–1376.1)	1140.9 (867.6–1238.9)	0.455
Phosphorous to protein ratio	24.6 (22.8–26.2)	25.1 (22.8–27.2)	0.356

Data are represented as median (Q1–Q3). *p* value was calculated by Mann Whiney U test. Table 5 indicates that the serum creatinine was statistically higher among male as compared to female (*p* < 0.001). * n = 29 (RPI) and n = 34 (HPhI). # n = 28 (RPI) and n = 31 (HPhI).

**Table 6 nutrients-14-03302-t006:** Univariable logistic regression analysis to identify the risk factors for CKD progression.

Parameters	Odds Ratio (OR)	95% Confidence Interval (CI)	*p*-Value
Age	1.179	0.411–3.376	0.760
Gender	0.935	0.386–2.267	0.882
Diet type (veg/non-veg)	2.600	1.027–6.585	0.044
Family history of CKD	3.841	1.499–9.839	0.005
Diabetes mellitus	0.762	0.296–1.962	0.573
Hypertension	0.084	0.018–0.397	0.002

**Table 7 nutrients-14-03302-t007:** Multivariable logistic regression to identify the risk factors for CKD progression.

Parameters	Adj. Odds Ratio (AOR)	95% Confidence Interval (CI)	*p*-Value
Diet type (veg/non-veg)	2.568	0.883–7.471	0.083
Hypertension	0.082	0.0.16–0.412	0.002
Family history of CKD	2.895	1.013–8.275	0.047

**Table 8 nutrients-14-03302-t008:** Comparison of biochemical parameters before and after dietary counselling and dietary intervention.

Parameters (Unit)	Groups	n	Baseline	At 6 Months	At 12 Months	*p*-Value
GFR (mL/min/1.73 m^2^)	HPhI	39	80.93 ± 15.34	84.11 ± 15.38	87.43 ± 18.27	0.012
	RPhI	34	85.19 ± 17.98	86.76 ± 18.96	87.45 ± 21.01	0.304
Serum phosphorous (mg/dL)	HPhI	39	3.95 ± 0.50	3.36 ± 0.78	3.36 ± 0.78	0.000
	RPhI	34	3.15 ± 0.57	3.25 ± 0.61	3.32 ± 0.64	0.177
iPTH (pg/mL)	HPhI	39	54.74 ± 10.41	57.54 ± 10.35	54.97 ± 11.42	0.097
	RPhI	34	48.68 ± 8.71	52.09 ± 7.67	50.69 ± 10.23	0.055
Serum creatinine (mg/dL)	HPhI	39	1.00 ± 0.21	0.96 ± 0.18	0.94 ± 0.22	0.102
	RPhI	34	0.96 ± 0.21	0.95 ± 0.22	0.95 ± 0.24	0.307
FGF-23 (pg/mL) Normal range: 8.2 to 54.3 pg/mL	HPhI	31	60.67 ± 6.26	58.00 ± 7.07	53.29 ± 9.48	0.000
	RPhI	30	53.70 ± 3.98	54.54 ± 6.84	53.07 ± 8.59	0.428
sKlotho (pg/mL) Normal range: 239 to 1266 pg/mL	HPhI	30	656.54 ± 51.26	815.35 ± 109.96	855.42 ± 163.29	0.000
	RPhI	26	800.58 ± 75.21	881.43 ± 169.18	891.37 ± 233.58	0.022
Urinary phosphorous (mg/day)	HPhI	24	574.37 ± 214.22	541.32 ± 218.35	624.64 ± 137.67	0.052
	RPhI	20	649.02 ± 185.07	638.16 ± 153.54	724.37 ± 128.15	0.071
Urinary protein (mg/mL)	HPhI	25	22.57 ± 3.09	23.61 ± 3.98	22.01 ± 3.39	0.071
	RPhI	20	16.79 ± 2.95	17.76 ± 2.91	16.42 ± 2.56	0.202
Serum calcium (mg/dL)	HPhI	37	8.69 ± 0.93	8.49 ± 1.46	8.68 ± 1.75	0.663
	RPhI	33	8.93 ± 0.53	9.17 ± 1.06	9.07 ± 1.47	0.518
Vitamin D (ng/mL)	HPhI	39	27.02 ± 8.55	24.75 ± 8.23	26.24 ± 7.74	0.394
	RPhI	28	26.91 ± 8.04	28.65 ± 8.07	28.05 ± 6.17	0.295
Dietary Protein intake (g/day)	HPhI	39	49.26 ± 5.45	43.55 ± 5.89	42.47 ± 6.65	0.001
	RPhI	34	37.87 ± 2.76	37.54 ± 4.25	38.54 ± 4.89	0.319
Dietary phosphorous intake (mg/day)	HPhI	39	1322.09 ± 132.66	1157.00 ± 156.25	1040.86 ± 191.36	0.000
	RPhI	34	871.88 ± 59.98	879.32 ± 82.76	905.23 ± 116.62	0.156
Phosphorous to protein ratio	HPhI	39	27.16 ± 4.35	26.94 ± 4.86	24.75 ± 4.34	0.001
	RPhI	34	23.11 ± 2.11	23.74 ± 3.61	23.84 ± 4.18	0.379

FGF-23: fibroblast growth factor-23; iPTH: intact parathyroid hormone; GFR: glomerular filtration rate. *p*-value calculated by repeated measures of ANOVA.

**Table 9 nutrients-14-03302-t009:** Comparison of biochemical parameters before and after dietary counselling and dietary intervention on the basis of gender.

Parameters (Unit)	Groups	Baseline	At 6 Months	At 12 Months	*p*-Value
GFR (mL/min/1.73 m^2^)	Male	85.8 (67.6–99.7)	94.47 (69.9–100.2)	91.04 (74.24–106.67)	0.004
	Female	90.1 (64–93.7)	83.28 (69.67–94.46)	80.99 (68.8–95.75)	0.342
Serum creatinine (mg/dL)	Male	1 (0.9–1.2)	0.98 (0.87–1.25)	0.96 (0.84–1.2)	0.004
	Female	0.8 (0.7–1.1)	0.85 (0.73–1.02)	0.85 (0.71–1)	0.15
Serum phosphorous (mg/dL)	Male	3.7 (3.2–4.1)	3.32 (2.78–3.97)	3.31 (2.87–4)	0.171
	Female	3.5 (2.8–3.8)	3.26 (2.62–3.7)	3.14 (2.72–3.72)	0.089
iPTH (pg/mL)	Male	51.5 (43.3–60.1)	55.03 (48.31–61)	51.02 (43.76–62.32)	0.003
	Female	51.8 (44.7–60.2)	55.05 (48.21–63.9)	52.42 (47.03–61.87)	0.005
FGF-23 (pg/mL)	Male	57.3 (53.6–61.6)	56.22 (51.81–58.84)	52.49 (46.98–60.43)	0.072
	Female	55.2 (52.2–59.1)	56.15 (51.44–61.62)	55.53 (49.33-59.1)	0.009
sKlotho (pg/mL)	Male	702.8 (648.7–811.7)	802.4 (734.73–914.98)	819.22 (682.73–1024.1)	<0.001
	Female	703.6 (673.9–840.7)	893 (751.91–983.19)	853.85 (762.54–1020.45)	<0.001
Urinary phosphorous (mg/day)	Male	661 (476.6–798.6)	597.1 (464.73–714.24)	692.37 (534.29–791.48)	0.136
	Female	638.9 (403.4–804.3)	585.5 (357.39–725.81)	699.7 (568.21–789.45)	0.071
Urinary protein (mg/mL)	Male	20.5 (18.3–22.5)	21.06 (18.42–25.56)	18.95 (16.91–22.19)	0.003
	Female	19.4 (15.8–22.5)	19.63 (16.45–22)	18.58 (15.82–22.18)	0.835
Serum calcium (mg/dL)	Male	8.9 (8.6–9.1)	9.09 (7.02–10.18)	8.5 (7.95–10.3)	0.645
	Female	9 (8.3–9.3)	8.95 (8.04–9.44)	8.8 (7.3–9.76)	0.795
Vitamin D (ng/mL)	Male	28.1 (19.8–32.4)	26.3 (19.43–30.98)	26.32 (21.88–30.12)	0.518
	Female	27.4 (18.7–30.8)	26.6 (23.7–32.4)	27.4 (24.38–33.09)	0.819
Dietary Protein intake	Male	43.7 (38.7–52.1)	40.89 (38.43–46.1)	41.21(38.44–45.32)	0.001
	Female	40.3 (36.6–47.9)	39.31 (34.75–43.41)	38.4 (35.08–43.1)	0.002
Dietary phosphorous intake (mg/day)	Male	1161.7 (883.5–1376.1)	1027.56 (905.01–1194.04)	957 (887.69–1095)	0.03
	Female	1140.9 (867.6–1238.9)	967.35 (824.56–1148.85)	871.73 (842–1003)	<0.001

*p* value was calculated by Friedman test.

## Data Availability

Data is available for review with the first author (AS).

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
