# Peer review of "Effect of Dietary Phosphorous Restriction on Fibroblast Growth 2 Factor-23 and sKlotho Levels in Patients with Stages 1–2 Chronic Kidney Disease"

_nutrients, 2022, doi:10.3390/nu14163302_

Round 1
Reviewer 1 Report
1. Why authors did not use a standardized CKD-EPI equation according to KDIGO recommendations? - please add explanation in the paper. Nevertheless, in line 82, please add which MDRD formula was used (short, long)?
2. Please provide what method to creatinine measurements was used?
3. Was the sample size calculated, as there is a great concern abot the power of the tests used in the paper. If no, please provide post-hoc power analysis.
4. What was the rationale to divide groups according to phosphorus intake < and >=1000 mg/day?
5. Why authors did not check the distributions of data?. There is a great concenr about missleading statistical analysis due to skeweness of data (such as f.e.: iPTH, urinary phosphorus, sKlotho). Values with skewed distribution should be presented with median and lower/upper quartiles.
6. Line 145 - p should be lowercase.
7. Please round age without decimals.
8. Significant p-values should be presented as p < 0.05, < 0.01 and < 0.001 in the whole paper.
9. In my opinion all analysis should be done taking sex into account.
10. Table 4 should be replaced by the multivariable linear or non-parametric regression.
11. Regarding the data decimals should be set appropriate to the measurement significance as f.e. Table 6 is hard to see.
Author Response
Reviewer 1 Queries 1- 11 Thank you for your queries and suggestions. Reply to the queries is given below:
- Why authors did not use a standardized CKD-EPI equation according to KDIGO recommendations? - please add explanation in the paper. Nevertheless, in line 82, please add which MDRD formula was used (short, long)?
Reply: Thank you for your comment. Short MDRD formula was used in place of CKD-EPI equation as the MDRD formula was recommended by experts in the research/doctoral committee hence we had to comply by the recommendations. Every project goes through scrutiny before it is submitted to the ethics committee and reviewed annually. The internal and external experts were nephrologists, immunologists.
- Please provide what method to creatinine measurements was used?
Reply: Jafee’s method was used for creatinine measurements.
- Was the sample size calculated, as there is a great concern about the power of the tests used in the paper. If no, please provide post-hoc power analysis.
Reply: Sample size was calculated for the project. However, because of word limit, the details on sample size calculation etc were withheld.
To find the association of FGF-23, Klotho, PTH levels with dietary phosphorus in CKD Stages 1 and 2 sample sizes was calculated in consultation with a bio-statistician. According to the null hypothesis, when r =0, there is no correlation. A negative correlation will be at r= 0.5 and positive >0.5 at a power of 90%. Accordingly, for this study, a minimum sample of 37 was sufficient to ascertain the association.
Hence, a total of 187 study subjects were enrolled from May 2016-December 2018 for the study out of which 81 were taken as controls and 83 as CKD patients who gave their consent for participating in the study. Out of 81 healthy controls. After drop outs a total of 79 CKD
Note: This project was reviewed by research/doctoral committee which ensures all the salient features like sample size, methodology etc which determine the quality of the work are fulfilled before submitting the project to the Ethics committee
- What was the rationale to divide groups according to phosphorus intake < and >=1000 mg/day?
Reply: The rationale to divide groups according to phosphorus intake <1000 and >=1000 mg/day was based on an average healthy Indian’s protein intake which is 0.6-0.7g/kg/d (equal to approximately 700-800 mg/d of phosphorus intake daily). Hence, the groups were divided based on KDOQI guidelines for phosphorus intake in CKD, which is 800-1000 mg/d (although in CKD stages 1 or 2, neither phosphorus restriction nor use of phosphate binders is recommended). (Reference: Salomo L, Kamper AL, Poulsen GM, Poulsen SK, Astrup A, Rix M. Habitual dietary phosphorus intake and urinary excretion in chronic kidney disease patients: a 3-day observational study. Eur J Clin Nutr. 2017;71(6):798-800.)
- Why authors did not check the distributions of data?. There is a great concenr about missleading statistical analysis due to skeweness of data (such as f.e.: iPTH, urinary phosphorus, sKlotho). Values with skewed distribution should be presented with median and lower/upper quartiles.
Reply : Thank you for your suggestion.
Normality of data was examined by Shapiro Wilk test. Results were presented as Mean ± SD for normally distributed while Median (IQR) for skewed data and Categorical as n (%). This project was reviewed by research/doctoral committee which ensures all the salient features like sample size, methodology etc which determine the quality of the work are fulfilled before submitting the project to the Ethics committee
- Line 145 - p should be lowercase.
Reply: Thank you for your suggestion. “P” changed to lowercase “p”
- Please round age without decimals.
Reply: Thank you for your suggestion. Rounded up age without decimals. Mean 38.00±12.06 Male 41.00±9.22 Female 35.00±13.91
- In my opinion all analysis should be done taking sex into account.
Reply: Analyzed data sex wise. Tables are given below:
Table 1b. Clinical characteristics of CKD patients (n=79).
|
|
|
Male(n=42) |
Female(n=37) |
|
|||
|
Diet type- Veg |
46 |
23 (29.11%) |
23 (29.11%) |
|
|||
|
non-veg |
33 |
19 (24.05%) |
14 (17.72%) |
|
|||
|
Diabetes, number (%) |
26 (32.9%) |
13 (16.4%) |
13 (16.4%) |
|
|||
|
Dyslipidemia |
15 (18.9%) |
6 (7.59%) |
9 (11.39%) |
|
|||
|
Hypertensive |
23 (29.1%) |
12 (15.18%) |
11 (13.92%) |
||||
|
Family history with CKD |
37 (46.8%) |
18 (22.78 %) |
19 (24.05%) |
|
|||
|
Mean age, years |
38.27±12.06 |
41.42±9.22 |
34.70±13.91 |
|
|||
|
Glomerular filtration rate, ml/min/1.73m2 |
83.69±17.37 |
85.10±17.95 |
82.08±16.78 |
|
|||
|
Renal diagnosis |
|
||||||
|
Diabetic kidney disease |
18 (22.7%) |
10 (12.65%) |
8 (10.12%) |
|
|||
|
Polycystic kidney disease |
14 (17.7%) |
6 (7.59%) |
8 (10.12%) |
|
|||
|
Other renal diseases |
16 (20.2 %) |
9 (11.39%) |
7 (8.86%) |
|
|||
|
Unknown |
31 (39.2 %) |
17 (21.51%) |
14 (17.72%) |
|
|||
|
Medications |
|
||||||
|
Anti-hypertensive drug(s) |
23 (29.1%) |
12 (15.18%) |
11 (13.92%) |
|
|||
|
Statins |
16 (20.2 %) |
7 (8.86%) |
9 (11.39%) |
|
|||
|
Insulin or oral hypoglycaemic agent |
26 (32.9%) |
13 (16.4%) |
13 (16.4%) |
|
|||
Values are presented in Mean±SD/n (%).CKD: Chronic Kidney Disease.
Table 3b. Demographic and baseline biochemical data of the CKD patients based on gender.
|
|
Male(n=42) |
Female(n=42) |
p value |
|
GFR |
85.8(67.6-99.7) |
90.1(64-93.7) |
0.415 |
|
Serum creatinine (mg/dL) |
1(0.9-1.2) |
0.8(0.7-1.1) |
<0.001 |
|
Serum phosphorous(mg/dL) |
3.7(3.2-4.1) |
3.5(2.8-3.8) |
0.374 |
|
iPTH (pg/mL) |
51.5(43.3-60.1) |
51.8(44.7-60.2) |
0.655 |
|
FGF-23 (pg/mL) |
57.3(53.6-61.6) |
55.2(52.2-59.1) |
0.196 |
|
sKlotho(pg/mL) |
702.8(648.7-811.7) |
703.6(673.9-840.7) |
0.482 |
|
Urinary phosphorous(mg/day)* |
661(476.6-798.6) |
638.9(403.4-804.3) |
0.448 |
|
Urinary protein (mg/mL)# |
20.5(18.3-22.5) |
19.4(15.8-22.5) |
0.124 |
|
Serum calcium(mg/dL) |
8.9(8.6-9.1) |
9(8.3-9.3) |
0.555 |
|
Vitamin D (ng/mL) |
28.1(19.8-32.4) |
27.4(18.7-30.8) |
0.613 |
|
Energy intake (kcal /kg/day) |
10.2(7.8-14.6) |
11(7.4-14.5) |
0.804 |
|
Dietary Protein intake (gm/ day) |
43.7(38.7-52.1) |
40.3(36.6-47.9) |
0.054 |
|
Dietary phosphorous (mg/day) |
1161.7(883.5-1376.1) |
1140.9(867.6-1238.9) |
0.455 |
|
Phosphorous to protein ratio |
24.6(22.8-26.2) |
25.1(22.8-27.2) |
0.356 |
p value was calculated by Mann Whiney U test
Table 3b indicates that the Serum Creatinine was statistically higher among male as compared to female (p<0.001).
Table 4b. Correlation of biochemical parameters with dietary phosphorous and phosphorous protein ratio in CKD patients among Male and Female.
|
Parameters(unit) |
‘r’ value, p-value |
|||
|
Male (n=42) |
Female (n=37) |
|||
|
Dietary Phosphorous |
P-P Ratio |
Dietary Phosphorous |
P-P Ratio |
|
|
GFR (ml/min/1.73m2) |
-0.239,0.128 |
0.886,<0.001 |
0.006,0.971 |
0.653,<0.001 |
|
Serum phosphorous(mg/dL) |
0.704,<0.001 |
-0.099,0.534 |
0.425,0.009 |
0.19,0.259 |
|
iPTH (pg/mL) |
0.353,0.022 |
0.577,<0.001 |
0.283,0.09 |
0.204,0.227 |
|
FGF-23 (pg/mL) |
0.6,<0.001 |
0.246,0.117 |
0.388,0.018 |
0.114,0.503 |
|
sKlotho(pg/mL) |
-0.639,<0.001 |
0.537,<0.001 |
-0.623,<0.001 |
0.092,0.589 |
|
Urinary phosphorous(mg/day)* |
0.168,0.342 |
-0.515,<0.001 |
-0.159,0.409 |
-0.287,0.085 |
|
Urinary protein (mg/mL)# |
0.602,<0.001 |
0.162,0.359 |
0.439,0.022 |
-0.12,0.536 |
|
Serum calcium(mg/dL) |
-0.023,0.884 |
0.574,0.001 |
-0.131,0.438 |
0.119,0.554 |
|
Vitamin D (ng/mL) |
0.226,0.15 |
0.077,0.63 |
0.159,0.348 |
-0.057,0.737 |
|
Dietary Protein intake (gm/ day) |
0.901,<0.001 |
0.164,0.3 |
0.616,<0.001 |
0.044,0.794 |
|
Phosphorous to protein ratio |
0.886,<0.001 |
- |
0.653,<0.001 |
- |
Table 5b. Multivariate logistic regression to identify the risk factors for CKD progression
|
Parameters |
Adj. Odds Ratio (AOR) |
95% Confidence Interval(CI) |
p-value
|
|
Diet type (veg/non-veg) |
2.146 |
1.04-4.43 |
0.039 |
|
Hypertension |
0.137 |
0.038-0.493 |
0.002 |
Table 6b. Comparison of biochemical parameters before and after dietary counselling and dietary intervention based on gender
|
Parameters(unit) |
Groups |
Baseline |
At 6 months |
At 12 months |
p-value |
|
GFR (ml/min/1.73m2) |
Male |
85.8(67.6-99.7) |
94.47(69.9-100.2) |
91.04(74.24-106.67) |
0.004 |
|
Female |
90.1(64-93.7) |
83.28(69.67-94.46) |
80.99(68.8-95.75) |
0.342 |
|
|
Serum creatinine (mg/dL) |
Male |
1(0.9-1.2) |
0.98(0.87-1.25) |
0.96(0.84-1.2) |
0.004 |
|
Female |
0.8(0.7-1.1) |
0.85(0.73-1.02) |
0.85(0.71-1) |
0.15 |
|
|
Serum phosphorous(mg/dL) |
Male |
3.7(3.2-4.1) |
3.32(2.78-3.97) |
3.31(2.87-4) |
0.171 |
|
Female |
3.5(2.8-3.8) |
3.26(2.62-3.7) |
3.14(2.72-3.72) |
0.089 |
|
|
iPTH (pg/mL) |
Male |
51.5(43.3-60.1) |
55.03(48.31-61) |
51.02(43.76-62.32) |
0.003 |
|
Female |
51.8(44.7-60.2) |
55.05(48.21-63.9) |
52.42(47.03-61.87) |
0.005 |
|
|
FGF-23 (pg/mL) |
Male |
57.3(53.6-61.6) |
56.22(51.81-58.84) |
52.49(46.98-60.43) |
0.072 |
|
Female |
55.2(52.2-59.1) |
56.15(51.44-61.62) |
55.53(49.33-59.1) |
0.009 |
|
|
sKlotho(pg/mL) |
Male |
702.8(648.7-811.7) |
802.4(734.73-914.98) |
819.22(682.73-1024.1) |
<0.001 |
|
Female |
703.6(673.9-840.7) |
893(751.91-983.19) |
853.85(762.54-1020.45) |
<0.001 |
|
|
Urinary phosphorous(mg/day) |
Male |
661(476.6-798.6) |
597.1(464.73-714.24) |
692.37(534.29-791.48) |
0.136 |
|
Female |
638.9(403.4-804.3) |
585.5(357.39-725.81) |
699.7(568.21-789.45) |
0.071 |
|
|
Urinary protein (mg/mL) |
Male |
20.5(18.3-22.5) |
21.06(18.42-25.56) |
18.95(16.91-22.19) |
0.003 |
|
Female |
19.4(15.8-22.5) |
19.63(16.45-22) |
18.58(15.82-22.18) |
0.835 |
|
|
Serum calcium (mg/dL) |
Male |
8.9(8.6-9.1) |
9.09(7.02-10.18) |
8.5(7.95-10.3) |
0.645 |
|
Female |
9(8.3-9.3) |
8.95(8.04-9.44) |
8.8(7.3-9.76) |
0.795 |
|
|
Vitamin D (ng/mL) |
Male |
28.1(19.8-32.4) |
26.3(19.43-30.98) |
26.32(21.88-30.12) |
0.518 |
|
Female |
27.4(18.7-30.8) |
26.6(23.7-32.4) |
27.4(24.38-33.09) |
0.819 |
|
|
Dietary Protein intake |
Male |
43.7(38.7-52.1) |
40.89(38.43-46.1) |
41.21(38.44-45.32) |
0.001 |
|
Female |
40.3(36.6-47.9) |
39.31(34.75-43.41) |
38.4(35.08-43.1) |
0.002 |
|
|
Dietary phosphorous intake (mg/day) |
Male |
1161.7(883.5-1376.1) |
1027.56(905.01-1194.04) |
957(887.69-1095) |
0.03 |
|
Female |
1140.9(867.6-1238.9) |
967.35(824.56-1148.85) |
871.73(842-1003) |
<0.001 |
p value was calculated by Friedman test
- Table 4 should be replaced by the multivariable linear or non-parametric regression.
Reply: Since reviewer 2 has not made any suggestions to this effect, the authors have added Multivariate logistic regression to identify the risk factors for CKD progression as Table 5b
Table 5b. Multivariate logistic regression to identify the risk factors for CKD progression
|
Parameters |
Adj. Odds Ratio (AOR) |
95% Confidence Interval(CI) |
p-value
|
|
Diet type (veg/non-veg) |
2.146 |
1.04-4.43 |
0.039 |
|
Hypertension |
0.137 |
0.038-0.493 |
0.002 |
- 11. Regarding the data decimals should be set appropriate to the measurement significance as f.e. Table 6 is hard to see.
Reply: Apoplogies for the inconvenience caused as the formatting of the table is totally lost. I am attaching Table 6 as a separate file as the formatting is getting lost in the manuscript.
|
Parameters(unit) |
Groups |
n |
Baseline |
At 6 months |
At 12 months |
p-value |
|
||||||
|
GFR (ml/min/1.73m2) |
HPhI |
39 |
80.93±15.34 |
84.11±15.38 |
87.43±18.27 |
0.012 |
|
||||||
|
RPhI |
34 |
85.19±17.98 |
86.76±18.96 |
87.45±21.01 |
0.304 |
|
|||||||
|
Serum phosphorous(mg/dL) |
HPhI |
39 |
3.95±0.50 |
3.36±0.78 |
3.36±0.78 |
0.000 |
|
||||||
|
RPhI |
34 |
3.15±0.57 |
3.25±0.61 |
3.32±0.64 |
0.177 |
|
|||||||
|
iPTH (pg/mL) |
HPhI |
39 |
54.74±10.41 |
57.54±10.35 |
54.97±11.42 |
0.097 |
|
||||||
|
RPhI |
34 |
48.68±8.71 |
52.09±7.67 |
50.69±10.23 |
0.055 |
|
|||||||
|
Serum creatinine (mg/dL) |
HPhI |
39 |
1.00±0.21 |
0.96±0.18 |
0.94±0.22 |
0.102 |
|
||||||
|
RPhI |
34 |
0.96±0.21 |
0.95±0.22 |
0.95±0.24 |
0.307 |
|
|||||||
|
FGF-23 (pg/mL) Normal range: 8.2 to 54.3 pg/mL |
HPhI |
31 |
60.67±6.26 |
58.00±7.07 |
53.29±9.48 |
0.000 |
|
||||||
|
RPhI |
30 |
53.70±3.98 |
54.54±6.84 |
53.07±8.59 |
0.428 |
|
|||||||
|
sKlotho(pg/mL) Normal range: 239 to 1266 pg/mL |
HPhI |
30 |
656.54±51.26 |
815.35±109.96 |
855.42±163.29 |
0.000 |
|
||||||
|
RPhI |
26 |
800.58±75.21 |
881.43±169.18 |
891.37±233.58 |
0.022 |
|
|||||||
|
Urinary phosphorous(mg/day) |
HPhI |
24 |
574.37±214.22 |
541.32±218.35 |
624.64±137.67 |
0.052 |
|
||||||
|
RPhI |
20 |
649.02±185.07 |
638.16±153.54 |
724.37±128.15 |
0.071 |
|
|||||||
|
Urinary protein (mg/mL) |
HPhI |
25 |
22.57±3.09 |
23.61±3.98 |
22.01±3.39 |
0.071 |
|
||||||
|
RPhI |
20 |
16.79±2.95 |
17.76±2.91 |
16.42±2.56 |
0.202 |
|
|||||||
|
Serum calcium (mg/dL) |
HPhI |
37 |
8.69±0.93 |
8.49±1.46 |
8.68±1.75 |
0.663 |
|
||||||
|
RPhI |
33 |
8.93±0.53 |
9.17±1.06 |
9.07±1.47 |
0.518 |
|
|||||||
|
Vitamin D (ng/mL) |
HPhI |
39 |
27.02±8.55 |
24.75±8.23 |
26.24±7.74 |
0.394 |
|
||||||
|
RPhI |
28 |
26.91±8.04 |
28.65±8.07 |
28.05±6.17 |
0.295 |
|
RPHI |
|
878.86±54.18 |
888.55±89.58 |
914.60±127.06 |
0.177 |
|
|
Dietary Protein intake (gm/ day) |
HPhI |
39 |
49.26±5.45 |
43.55±5.89 |
42.47±6.65 |
0.000 |
|
RPHI |
|
878.86±54.18 |
888.55±89.58 |
914.60±127.06 |
0.177 |
|
RPhI |
34 |
37.87±2.76 |
37.54±4.25 |
38.54±4.89 |
0.319 |
|
RPHI |
|
878.86±54.18 |
888.55±89.58 |
914.60±127.06 |
0.177 |
|
|
Dietary phosphorous intake (mg/day) |
HPhI |
39 |
1322.09±132.66 |
1157.00±156.25 |
1040.86±191.36 |
0.000 |
|
RPHI |
|
878.86±54.18 |
888.55±89.58 |
914.60±127.06 |
0.177 |
|
RPhI |
34 |
871.88±59.98 |
879.32±82.76 |
905.23±116.62 |
0.156 |
|
RPHI |
|
878.86±54.18 |
888.55±89.58 |
914.60±127.06 |
0.177 |
|
|
Phosphorous to protein ratio
|
HPhI |
39 |
27.16±4.35 |
26.94±4.86 |
24.75±4.34 |
0.000 |
|
|
|
1317.51±137.20 |
1229.29±151.87 |
1137.48±172.15 |
0.00 |
|
RPhI |
34 |
23.11±2.11 |
23.74±3.61 |
23.84±4.18 |
0.379 |
|
RPHI |
|
|
|
|
|
|

Reviewer 2 Report
Hyperphosphatemia has emerged as an independent risk factor for cardiovascular disease (CVD) and excess mortality in chronic kidney disease (CKD). Using a one-year interventional study conducted on a 79 stage 1 and 2 CKD patients, Anita et. al found that dietary phosphorous restriction (<1g/d) improves renal function and attenuates hyperphosphatemia.
Major concerns
1. Dietary protein is known to affect CKD. However, patients in DPHI
and RPI groups have varied protein intake (Table 3). As such, how they can state that the improvement of renal function and hyperphosphatemia is due to phosphorous restriction.
2. It would be nice showing the reason for the sample size and also showing the data with Mean±SEM.
3. Dietary Protein intake (gm/ day) is assumed to be Dietary Protein intake (g/ day)?
4. In addition to high phosphorous load, are these any other dietary components (either nutrients or non-nutrients like phytochemicals) that can affect FGF-23 levels? This is important because patients between DPHI and RPI groups have many differences in respects of dietary foods.
5. Table 6 should be shown in one page.
6. How about the title "Effect of dietary phosphorous restriction on fibroblast growth 2 factor-23 and sKlotho levels in patients with stage 1-2 chronic kidney disease.
Author Response
Reviewer 2 Thank you for your queries and comments and suggestions.
Reply to Queries 1-6 is given below:
Major concerns
- Dietary protein is known to affect CKD. However, patients in DPHI and RPI groups have varied protein intake (Table 3). As such, how they can state that the improvement of renal function and hyperphosphatemia is due to phosphorous restriction.
Reply Table 3a.is h=showing the baseline biochemical data of the CKD patients based on their dietary phosphorous intake (mg/day). The infererence is based on table
Tab Table 6a.Comparison of biochemical parameters before and after dietary counselling and dietary intervention.
|
Parameters(unit) |
Groups |
n |
Baseline |
At 6 months |
At 12 months |
p-value |
|
Serum phosphorous(mg/dL) |
HPHI |
39 |
3.95±0.50 |
3.36±0.78 |
3.36±0.78 |
0.001 |
|
RPI |
34 |
3.15±0.57 |
3.25±0.61 |
3.32±0.64 |
0.177 |
|
|
GFR (ml/min/1.73m2) |
HPHI |
39 |
80.93±15.34 |
84.11±15.38 |
87.43±18.27 |
0.012 |
|
RPI |
34 |
85.19±17.98 |
86.76±18.96 |
87.45±21.01 |
0.304 |
- It would be nice showing the reason for the sample size and also showing the data with Mean±SEM.
Reply: Sample size was calculated for the project. However, because of word limit, the details on sample size calculation etc were withheld.
To find the association of FGF-23, Klotho, PTH levels with dietary phosphorus in CKD Stages 1 and 2 sample sizes was calculated in consultation with a bio-statistician. According to the null hypothesis, when r =0, there is no correlation. A negative correlation will be at r= 0.5 and positive >0.5 at a power of 90%. Accordingly, for this study, a minimum sample of 37 was sufficient to ascertain the association.
Hence, a total of 187 study subjects were enrolled from May 2016-December 2018 for the study out of which 81 were taken as controls and 83 as CKD patients who gave their consent for participating in the study. Out of 81 healthy controls. After drop outs a total of 79 CKD
We have represented data as Mean+/- SD as it describes the dispersion of data points about mean which helps in under standing the distribution of data whereas Mean+/- SEM describes the sampling distribution of sample mean.
Note: This project was reviewed by research/doctoral committee which ensures all the salient features like sample size, methodology etc which determine the quality of the work are fulfilled before submitting the project to the Ethics committee
- Dietary Protein intake (gm/ day) is assumed to be Dietary Protein intake (g/ day)?
Reply: Thank you for your suggestion Protein intake is g/ day and it has been corrected in the manuscript.
- In addition to high phosphorous load, are these any other dietary components (either nutrients or non-nutrients like phytochemicals) that can affect FGF-23 levels? This is important because patients between DPHI and RPI groups have many differences in respects of dietary foods.
Reply: Thank you for your suggestion. Yes, phyotochemicals and antioxidants do have a bearing on phosphate metabolism, but this aspect was not examined. This statement has been included in methodology.
- Table 6 should be shown in one page.
Reply: Thank you for your comment. Table 6 is losing formatting therefore table 6 is attched as a separate file.
- How about the title "Effect of dietary phosphorous restriction on fibroblast growth 2 factor-23 and sKlotho levels in patients with stage 1-2 chronic kidney disease.
Reply: I respect the experience of the reviewer and his/her suggestion to improve the title.. If the reviewer has suggested the title” "Effect of dietary phosphorous restriction on fibroblast growth 2 factor-23 and sKlotho levels in patients with stage 1-2 chronic kidney disease.” I would not mind improvising the title to "Effect of dietary phosphorous restriction on fibroblast growth 2 factor-23 and sKlotho levels in patients with stage 1-2 chronic kidney disease. as suggested by the reviewer 2.

Round 2
Reviewer 1 Report
1. Please provide the sample size calculation, otherwise the power of the significant results.
2. Tables - please change the number of decimal places accordingly to its measurements (it is wrong to show age 38.00 with sd = 12.06, should be rounded to whole numbers, GFR rounded to 1 decimal place, and so on).
3. Table 1 - please provide % of male/female and diet type. For DM male lack of %).
4. Tables - there should be spaces between +- sign and numbers as well as between numbers and parenthesis.
5. Table 1, lack of info below table about mean/sd/median/quartiles.
6. Table 2, lack of info below table about median/quartiles. It should be started with upper case S.
7. Table 4a/4b should be removed and proper multivariable analyses should be done. Based on this conclusions should be made.
8. Instead multivariate use multivariable.
9. Table 6a - change significant p-values accordingly to its presentation in the text.
Author Response
All the queries raised have been answered and incorporated in the manuscript.
- Please provide the sample size calculation, otherwise the power of the significant results.
Sample size: To find association of FGF-23, Klotho, PTH levels with dietary Phosphorus in Chronic Kidney Disease Stages 1 and 2 sample size was calculated in consultation with Head department of Biostatistics SGPGIMS. . According to null hypothesis, when r =0, there is no correlation. Negative Correlation will be at r= 0.5 and positive >0.5 at power of 90% . Accordingly a minimum sample of 73 and not 37 (apologies for typing error which was overlooked over and over again despite being reminded) would suffice to ascertain the association.
z tests - Correlations: Two independent Pearson r's Analysis:
A priori: Compute required sample size Input: Tail(s) = Two
Effect size q = -0.5493061
α err prob = 0.05
Power (1-β err prob) = 0.90
Allocation ratio N2/N1 = 1
Output: Critical z = -1.9599640
Sample size group 1 = 73
Sample size group 2 = 73
Total sample size = 146
Actual power = 0.9014357
Please Note :total sample size both intervention and control were 73 each. However in this manuscript only CKD patients (n 79 ) have been included. The calculations are given here but not included in the manuscript. The sample size has been corrected.
- Tables - please change the number of decimal places accordingly to its measurements (it is wrong to show age 38.00 with sd = 12.06, should be rounded to whole numbers, GFR rounded to 1 decimal place, and so on).
Reply Thankyou for your comment. WE have done the needful
|
Diet type- Veg Male Female non-veg Male Female
|
46 23 (50%)*23 (50%)* 33 19(57.6%)* 14(42.4%)* |
|
Mean age, years Male Female |
38.±12.† 41.±9.† 35.±13† |
- Table 1 - please provide % of male/female and diet type. For DM male lack of %).
|
Diabetes, number (%) Male |
26 (32.9%)* 13 (16.4%)* |
Please Note: a lot of information was lost in the Tables because of changes in the formatting
- Tables - there should be spaces between +- sign and numbers as well as between numbers and parenthesis.
Reply Thankyou for your comment. WE have done the needful
- Table 1, lack of info below table about mean/sd/median/quartiles.
Reply Thankyou for your comment. WE have done the needful for all the Tables
Table 2 *n=63#n=59FGF-23: Fibroblast Growth Factor-23; iPTH: Intact Parathyroid Hormone Values are presented in $Mean ± SD and †Median (Q1 – Q3)
Table 3 a *n=29 (RPI) and n=34 (HPhI)# n=28 (RPI) and n=31 (HPhI)FGF-23: Fibroblast Growth Factor-23; iPTH: Intact Parathyroid Hormone; GFR: Glomerular Filtration RateValues are presented in Mean .± SD and Median (Q1 – Q3).
@ p value was calculated using independent sample t-test; $ p value was calculated by Mann Whiney U test;†p-value calculated by chi-square test
Table 3 b Data is represented as Median (Q1 – Q3)
p value was calculated by Mann Whiney U test
Table 3b indicates that the serum creatinine was statistically higher among male as compared to female (p<0.001).
Table 4a Univariable logistic regression analysis showed diet type, family history and hypertension as risk factors for CKD progression in CKD (Table 4a).
Multivariable logistic regression analysis was performed to find out the risk factors for CKD progression based on phosphorus intake of CKD patients. Risk factors which had a statistical bearing on the progression of CKD were, hypertension (AOR= 0.082, p value=0.002) and family history of CKD (AOR = 2.895, p value = 0.047).
Table 5 Comparison of biochemical parameters before and after dietary counselling and dietary intervention. (On deleting previous Table 4 Table 5 is now table 4 and Table 6 is now Table 5)
Table 1 b Values are presented in Mean±SD/n (%).CKD: Chronic Kidney Disease
Table 1 a Values are presented in Mean ± SD/n (%).CKD: Chronic Kidney Disease
- Table 2, lack of info below table about median/quartiles. It should be started with upper case S.
Reply Thankyou for your comment. We have done the needful
*n=63#n=59FGF-23: Fibroblast Growth Factor-23; iPTH: Intact Parathyroid Hormone Values are presented in $Mean ± SD and †Median (Q1 – Q3)
- Table 4a/4b should be removed and proper multivariable analyses should be done. Based on this conclusions should be made.
Reply Thankyou for your comment. Table 4a nd 4b have been deleted
The findings of the study emphasize that in CKD patients, on predominantly animal-based diets, there is a need for cautious dietary phosphorus intake from early stages of CKD as the univariable analysis has shown dietary phosphorus as a risk factor for CKD progression. However in multivariable analysis the dietary phosphoorus consumption did not appear as a risk factor for CKD progression , rather only family history of CKD and hyperrension were risk factor for CKD
Tables 5 a and b are Tables 4 a and b
Table 4a Univariable logistic regression analysis showed diet type, family history and hypertension as risk factors for CKD progression in CKD (Table 4a).
Multivariable logistic regression analysis was performed to find out the risk factors for CKD progression based on phosphorus intake of CKD patients. Risk factors which had a statistical bearing on the progression of CKD were,hypertension (AOR= 0.082, p value=0.002) and family history of CKD (AOR = 2.895, p value = 0.047).
Discussion: Most of our CKD patients who were non-vegetarian; were in the higherHPhI group, this being responsible for their high phosphorus intake. The findings of the study emphasize that in CKD patients, on predominantly animal-based diets, there is a need for cautious dietary phosphorus intake from early stages of CKD as the univariable analysis has shown dietary phosphorus as a risk factor for CKD progression. However in multivariable analysis the dietary phosphoorus consumption did not appear as a risk factor for CKD progression , rather only family history of CKD and hyperrension were risk factor for CKD progression.While taking the dietary recall, the renal dietician should educate and counsel CKD patient on the sources of phosphorus, and type of diet the patient should prefer. Also, the importance of low phosphorus–to-protein-ratio diet should be emphasized in dietary counselling and educating patients with CKD.27
- Instead multivariate use multivariable.
Reply Thankyou for your comment. We have used the term “multivariable.”
- Table 6a - change significant p-values accordingly to its presentation in the text.
Reply Thankyou for your comment.. Previous Table 6ais table 5a now. We have done the needful. Please Note: Table 5a is being sent to as a separate file as the in the manuscript the formatting is getting lost (is not maintained)
